# Stereo Vision for Plant Detection in Dense Scenes

**DOI:** 10.3390/s24061942

**Published:** 2024-03-18

**Authors:** Thijs Ruigrok, Eldert J. van Henten, Gert Kootstra

**Affiliations:** Farm Technology, Department of Plant Sciences, Wageningen University and Research, 6700 AA Wageningen, The Netherlands

**Keywords:** vegetation density, stereo vision, multimodal, deep learning, precision weed control

## Abstract

Automated precision weed control requires visual methods to discriminate between crops and weeds. State-of-the-art plant detection methods fail to reliably detect weeds, especially in dense and occluded scenes. In the past, using hand-crafted detection models, both color (RGB) and depth (D) data were used for plant detection in dense scenes. Remarkably, the combination of color and depth data is not widely used in current deep learning-based vision systems in agriculture. Therefore, we collected an RGB-D dataset using a stereo vision camera. The dataset contains sugar beet crops in multiple growth stages with a varying weed densities. This dataset was made publicly available and was used to evaluate two novel plant detection models, the D-model, using the depth data as the input, and the CD-model, using both the color and depth data as inputs. For ease of use, for existing 2D deep learning architectures, the depth data were transformed into a 2D image using color encoding. As a reference model, the C-model, which uses only color data as the input, was included. The limited availability of suitable training data for depth images demands the use of data augmentation and transfer learning. Using our three detection models, we studied the effectiveness of data augmentation and transfer learning for depth data transformed to 2D images. It was found that geometric data augmentation and transfer learning were equally effective for both the reference model and the novel models using the depth data. This demonstrates that combining color-encoded depth data with geometric data augmentation and transfer learning can improve the RGB-D detection model. However, when testing our detection models on the use case of volunteer potato detection in sugar beet farming, it was found that the addition of depth data did not improve plant detection at high vegetation densities.

## 1. Introduction

Conventional agriculture relies on herbicides to prevent infestations by weeds [1,2]. Currently, society and politics encourage more sustainable production methods and are pushing towards organic agriculture [3], which is currently labor intensive. Automated precision weed-control is a promising solution to reduce herbicide usage while maintaining a high yield and low labor demand. To realize automated precision weed control, a visual method to discriminate between crops and weeds (further referred to as a plant detection method) is needed. 

Current state-of-the-art plant detection methods operate using 2D images and rely on deep learning-based object detection algorithms such as YOLO [4], SSD [5], and Mask-RCNN [6]. These models, as introduced in their respective seminal papers, have been applied successfully in weed detection studies [7,8,9,10,11,12]. These models are typically trained on large datasets, such as the Microsoft Common Objects in Context (MS-COCO) dataset [13], and subsequently fine-tuned on domain-specific weed detection datasets using transfer learning and data augmentation techniques. This approach enhances model robustness by exploiting generic features learned from publicly available datasets. These modern methods have demonstrated superior effectiveness and robustness compared to older feature-based detection models. However, their performance remains unsatisfactory in dense and occluded scenes where weeds are difficult to detect reliably [12,14,15]. 

The difficulty in detecting weeds in dense and occluded scenes is not a new problem. Handcrafted feature models have faced similar challenges in the past. To overcome this limitation of the handcrafted algorithms, researchers have successfully utilized depth data to improve plant detection accuracy in dense and occluded scenes [16,17]. With the increasing availability, quality, and use of stereo cameras in agriculture [18,19,20,21,22], it is surprising that depth data is not commonly used for deep learning-based vision systems for weed detection in agriculture.

In a literature review, Kurtser and Lowry [23] identified 24 papers that applied a combination of both color and depth data (RGB-D data) for robotic perception in precision agriculture. Almost all these papers focused on object detection solely using RGB images. Depth data were primarily utilized for pre- or post-processing steps, such as background filtering, or for performing measurements based on geometric properties such as crop volume [19]. Among the analyzed papers, only Gené-Mola et al. [24] employed a combination of color, depth, and infrared data for the detection task itself. Their results indicated that the addition of both infrared and depth data substantially improves the detection performance. However, they did not specifically assess the impact of depth data on detection performance. Our study contributes to this emerging field by studying the efficacy of combining color and depth images for plant detection.

One of the main reasons that RGB-D data are not frequently used for object detection in agriculture is the lack of large general RGB-D datasets to pretrain deep neural networks [23]. Therefore, it is important to investigate other methods that can facilitate better training of RGB-D models, such as data augmentation and transfer learning from the RGB domain. However, data augmentation [25,26,27,28,29,30] and transfer learning [31,32], commonly used in state-of-the-art RGB detection models, are not one-to-one applicable to RGB-D data. This hampers the training of deep networks for RGB-D data.

A potential way to overcome this challenge is to apply a procedure proposed and successfully implemented in a household object classification method proposed by Eitel et al. [33]. In their paper, the authors proposed an effective and computationally inexpensive method: encoding of depth data as a color image. First, the depth data were normalized, and then a jet colormap was applied. This color map mapped the depth data to color values ranging from red (near) to green to blue (far), essentially distributing the depth information over all three RGB channels. Colorizing the depth data and representing them as an color image allowed them to apply existing 2D deep learning models as well as common augmentation and transfer learning techniques to boost detection performance. 

Our study focuses on exploring the potential of RGB-D data in combination with deep learning object detection models to perform plant detection in dense and occluded scenes. By combining depth data with color data in a deep learning object detection model, we aim to improve the accuracy and reliability of weed detection, particularly in dense and occluded scenes, where traditional RGB-based approaches fall short. To explore the potential of RGB-D data for plant detection in dense scenes, we developed two new deep learning plant detection models: the D-model, using depth data as the input, and the CD-model, using both color and depth data as inputs. As a reference model, the C-model, which uses only color images as the input and represents the current state-of-the-art in plant detection, was included. After suitable transformations were applied to the RGB-D data, we studied the effects of conventional data augmentation techniques and transfer learning methods on the new D- and CD-model. Furthermore, we investigated whether the usage of RGB-D data could improve plant detection in dense scenes compared to the use of only RGB data. We present a dataset including 1699 stereo pairs containing a total of 27,483 sugar beet and 2920 potato plants in dense cluttered scenes in various stages of development. We use this dataset to answer the following three research questions:To what extent does geometric and spectral data augmentation improve the performance of a detection model using both RGB and color-encoded depth data?Is transfer learning effective when transferring pre-trained RGB features to a model using RGB and color-encoded depth data?Can the combination of RGB and color-encoded depth data improve plant detection accuracy in dense and occluded scenes compared to using only color images?

Alongside this paper, we made our RGB-D dataset available to help the field progress. 

These questions were answered based on the detection of volunteer potato plants in sugar beet fields, a practical challenge in agricultural automation requiring visual discrimination between volunteer potato plants (weeds) and sugar beet plants (crops). Volunteer potato plants are regrown from tubers left in the soil after harvest. These volunteer potato plants compete with other crops for nutrients and water and can serve as hosts for pests and diseases that can infect other potato crops, resulting in economic losses for farmers. Due to the lack of chemical herbicides that kill potato plants effectively, an automated plant-specific removal method is needed [9,10,32,34]. Given the irregular growth pattern of volunteer potato plants and their capacity to regenerate from tubers even after the removal of above-ground parts, control efforts are essential throughout the entire growing season of the sugar beet crops. This includes the early stages after crop sprouting and extends into the later stages when the sugar beet plants have developed significant vegetation, leading to increased density and occlusion. Therefore, a volunteer potato detection system must be robust and effective, even in scenarios with heavy occlusion and a high vegetation density. 

## 2. Materials and Methods

In Section 2.1, we describe the image data set used for our research. Section 2.2 describes the detection models used, and finally, Section 2.3 describes the experiments performed to answer our three research questions.

### 2.1. Dataset Description

In this section, the image dataset used to train and test the detection model is described. In Section 2.1.1, the stereo camera system used to acquire the image dataset is presented. In Section 2.1.2, we explain how we calibrated the stereo camera and computed the depth map from the stereo image pairs. Finally, in Section 2.1.3, we explain when and where the images were acquired and how we divided the data into training and test datasets.

#### 2.1.1. Camera System

Data acquisition was performed using a stereo camera setup mounted to the hitch of a tractor, shown in Figure 1. As a stereo camera system, two IDS GV-5280FA-C-HQ cameras with a resolution of 2448 × 2048 were used. The cameras were equipped with a RICOH FL-CC0814-5M lens, resulting in a viewing angle of 57.8°. The cameras were mounted 20 cm apart with a vergence angle of 14° to maximize the overlap in the field of view between the cameras. The cameras were synchronized by a hardware trigger signal, and the shutter time was kept below 500 µs to prevent motion blur.

The camera system was mounted to the tractor hitch approximately 1.5 m above the ground, resulting in a field of view of approximately 1.3 m × 1.1 m and a spatial resolution of approximately 0.5 mm per pixel. During image acquisition, the tractor drove at a velocity of 0.7 m/s. An image was taken every half second, resulting in an image overlap of approximately 70% in the driving direction. 

It must be noted that the camera system did not contain a cover or artificial lighting. All the images were therefore acquired under natural illumination and were influenced by the weather conditions during the acquisition period. 

#### 2.1.2. Disparity Calculation and Color Encoding of Depth Data

The stereo camera was calibrated using the MATLAB 2018b stereovision toolbox [35]. The intrinsic and extrinsic parameters of the camera system were computed based on 178 image pairs of the MATLAB 2018b checkerboard calibration plate. Using the intrinsic and extrinsic parameters of the camera system, a disparity map of each image pair was calculated. The disparity calculation process consisted of three steps. In the first step, both the left and right images were rectified and aligned with each other. Secondly, the images were converted to grayscale. Thirdly, the disparity was calculated using the “disparity” function in MATLAB. In this function, stereo matching was performed using semi-global matching [36] with a block size of 5 pixels, a contrast threshold of 0.2, and a uniqueness threshold of 5. Afterward, the disparity maps were color-encoded using the MATLAB 2018b jet colormap. Using this jet color scheme, the disparity ranges were mapped to color values ranging from blue, representing a low/negative disparity, to green to red, representing a high disparity. By using this color encoding, the disparity data were distributed over three channels, similar to an RGB image. Since the detection model is designed for RGB images, the color-encoding procedure provides a common structure between disparity data and an RGB image [33]. In a preliminary study, we found that a detection model using jet color encoding outperformed a detection model using a single-channel disparity representation. Therefore, in this study, we used jet color encoding as a representation of depth in all our experiments. An example of the rectified images and the corresponding color-encoded depth map can be seen in Figure 2. 

#### 2.1.3. Data Acquisition 

The images were acquired in 2019 in three neighboring sugar beet fields located in Wageningen in The Netherlands. The fields were sown conventionally, with an inter-row spacing of 50 cm and an intra-row spacing of 20 cm. Potatoes were the preceding crop; therefore, volunteer potatoes occurred abundantly in the field. Finally, contrary to conventional practice, no weed management was applied; therefore, severe infestation of natural weeds occurred.

To ensure independent training, validation, and testing datasets, we assigned the images from the first field to the training dataset, the images from the second field to the validation dataset, and the images from the third field to the testing dataset. Image acquisition started on 22 May 2019 for the training and validation dataset and 23 May 2019 for the testing dataset. During the first data acquisition runs, the sugar beets were in the four-leaf stage. Furthermore, additional data were collected on 23 May 2019, 3 June 2019, and finally on 6 June 2019 when canopy closure occurred.

Afterwards, a selection of the data was made. First, the images acquired on the headland were removed, and only the images containing data showing the crop row were kept. Afterwards, a random sample of images was taken from the training and validation datasets. From this sample, only the images containing at least one potato plant were kept. In these final images, all the sugar beet and potato plants were manually annotated. In total, 865 annotated images were acquired for the training dataset, containing 12,754 sugar beet annotations and 1427 potato annotations. The validation dataset consisted of 440 annotated images, containing 7466 sugar beet annotations and 768 potato annotations. An overview of the data acquired on each day in each field is given in Table 1.

For the testing dataset, the data were sampled differently. The data were sorted based on vegetation density, which was defined as the ratio of green pixels in the image. The data were split into nine categories: 0–10%, 10–20%, and up to 80–90% vegetation density. Based on each vegetation density category, 200 images were selected, and only the images containing at least one potato plant were kept and annotated. An overview of the test dataset is given in Table 2.

### 2.2. Model Description

In this section, the detection models used are described. In Section 2.2.1, we explain why we used the YOLOv5L detection model, and we briefly describe its architecture. In Section 2.2.2, we explain how we modified the YOLOv5L detection model to process not only color but also depth data. How we applied data augmentation to the training data is explained in Section 2.2.3. Finally, in Section 2.2.4, we explain how we applied transfer learning to our detection models. 

#### 2.2.1. Model Description

Considering the use case of an automated volunteer potato removal robot, an accurate detection algorithm that runs in real-time was needed. The most commonly used state-of-the-art object detectors are Faster RCNN [37], RetinaNet [38], Detectron2 [39]), and YOLOv5 [40]. Of these object detectors, YOLOv5 is the fastest, with minor concessions on accuracy [4,12,40,41,42,43]. The YOLOv5 series consists of five models: N (nano), S (small), M (medium), L (large), and X (extra-large), with the smaller models optimized for speed and the larger models focused on achieving a higher detection accuracy. Of these models, we used the YOLOv5L model, the second-largest model. A performance increase of 1.7 mean average precision (mAP) for the MS-COCO dataset could be achieved using the extra-large YOLOv5X model. However, the YOLOv5X model requires double the computational time, which does not fit for its real-time application for precision weed control. 

The YOLOv5L design is mainly based on [4,41,42,43,44]. YOLOv5L is a single-shot multi-box detection method, meaning that a single pass through the model yields all bounding box predictions and object classifications of that image. This setup makes YOLOv5L faster than multi-stage approaches such as Faster RCNN [37] and Detectron2 [39], at the cost of a small sacrifice in accuracy. 

YOLOv5L uses a feature pyramid structure to create low-, medium-, and high-resolution feature maps, which are then used for the detection of objects in their respective scales. The detection model uses three anchor boxes per detection scale. Before training, the anchor box dimensions were optimized for the bounding-box dimensions in our training dataset.

All the models were trained using the hyper-parameters and Python library versions described by Jocher et al. [40] unless specified otherwise. An exploratory study showed that the models fully converged on our dataset after 100 epochs. Consequently, during all our experiments, the models were trained for 100 epochs. All training and evaluation was performed using a desktop PC equipped with an Intel Xeon E5-1603 processor, 64 Gb of RAM, and an Nvidia GeForce 1080 Ti GPU. 

#### 2.2.2. Model Modifications

To test the added value of depth information on the use case of weed detection for precision weed control, we defined three models, using color (C), depth (D), and color and depth (CD) as the input data. The C-model used only the three-channel RGB color image from the right camera as the input. This model represents the current state-of-the-art in plant detection and is therefore used as a baseline model. Of the two novel models, the D-model used only the three-channel color-encoded depth map as the input, and the CD-model used both the RGB color image and the color-encoded depth map as inputs. Of these three models, the C- and the D-model consisted of the standard YOLOv5L architecture, with three input channels. The CD-model consisted of a modified version of the YOLOv5L model that allowed for the usage of a color image and a depth map as inputs simultaneously. 

To allow the usage of color and depth data in the CD-model, a data fusion strategy was needed. The fusion strategies reported in the literature can be divided into three main strategies: decision-level fusion, layer-level fusion, and input-level fusion [45]. Of these three fusion strategies, we have chosen to implement the input-level fusion strategy, because this strategy has the lowest computational demand and the potential to learn relations between the different sensor modalities in a range of cases [45,46].

For the CD-model, the input-level fusion approach was implemented, similar to [46]. In the data pre-processing step, the channels of the color and depth images were concatenated, creating a six-channel input. Furthermore, the input layer of the YOLOv5L model was modified from 64 filters with three input channels each to 64 filters with six input channels each to allow for the processing of data with six channels.

#### 2.2.3. Data Augmentation

During the training time, data augmentation was applied to the training images. Each time, before an image was presented to the network, data augmentation was applied to the image. Because of the data augmentation, the network was never presented with the exact same image twice, creating a virtually infinite dataset. The data augmentation used in this research consisted of geometric and spectral data augmentation. Using geometric augmentation, the images were, for instance, flipped, cropped, and rotated, and using spectral augmentation, transformations in color space, such as changes in hue, brightness, and contrast, were applied. In the next two sections, we describe the geometric and spectral data augmentation process used. 

##### Geometric Data Augmentation

As geometric data augmentation, rotation, translation, scale, shear, vertical flip, and horizontal flip were implemented. The parameters for these augmentation steps were derived from the parameters provided by Jocher et al., [40]. To better fit the data distribution of our use case, we modified some data augmentation parameters. Due to the top-down view of the plants, it was considered realistic to rotate or flip all the plants in every direction. Therefore, we increased the rotation range to 360 degrees and allowed for image flipping in both the horizontal and vertical directions. Furthermore, we lowered the range of the scale parameter. Originally, the changes in scale were applied to a range of up to a 90% increase or decrease of the image dimensions. Because all the images were acquired from approximately the same distance from the plants, limited variations in the scale were present in our dataset. Hence, we lowered the scaling range to 50% of the image dimensions. Finally, we increased the shear transformation from 0 to 1 degree vertically and horizontally. The shear transformation influences the shape of the plants in the image, and applying a large amount of shear can make the plants look unrealistic. However, a small amount of shear can make the model more robust to small changes in shape. A list of the geometric data augmentation parameters is given in Table 3, and an example of the augmented data is provided in Figure 3. Note that the color image and the depth image both received the exact same transformation to maintain their alignment.

To match the annotations to the augmented image, the same transformation was applied to the annotations as well. The translation, scale, and flip augmentations could be directly applied to the annotations without sacrificing their quality. However, the rotation and shear transformation could not directly be applied to the annotations because the bounding boxes had their sides parallel to the sides of the image. Therefore, after applying the shear and rotation transformation to the annotation, a new bounding box was generated, inscribing the transformed annotation. This method ensured that the object of interest was still inscribed in the new bounding box. However, this also caused the modified bounding boxes to be larger than the original bounding boxes, which might reduce the accuracy of the localization. An example of the geometric data augmentation and the modified bounding boxes is shown in Figure 3.

##### Spectral Data Augmentation

Using spectral data augmentation, only transformations in hue, saturation, and value space (HSV) were applied. Spectral data augmentation does not require transformations of the annotations because it does not change the location or class of the objects. We did not apply any changes to the HSV augmentation parameters provided by Jocher et al. [40]. A list of the spectral data augmentation parameters used is given in Table 4. Instead of applying spectral data augmentation only to the RGB image, we also applied spectral data augmentation to the color-encoded depth image. Though this might yield some unrealistic pixel values that could not be achieved using depth calculations in the real world, it potentially improves the regularization of the detection model. An example of the augmented data is given in Figure 4.

#### 2.2.4. Transfer Learning

In this research, we used two methods to train the detection model: one without transfer learning and one with transfer learning. When no transfer learning was applied, the detection model was initialized with random weights/parameters and then trained on the available data consisting of 865 training images. When transfer learning was applied, the model was pre-trained on the MS-COCO dataset. This dataset contained more than 300,000 images from multiple domains and more than 1,500,000 annotated objects from 80 categories. Because of the sheer size of the MS-COCO dataset, the model was forced to learn generic features that are robust to noise and contain high generalization properties. When using transfer learning, we started with the model pre-trained on MS-COCO and fine-tuned this model on our 865 training images. When starting with a model that already “knows” generic features, the model is expected to converge faster and achieve a better generalization performance. 

To apply transfer learning to the C- and D-model, we started with the weights that were pretrained for 300 epochs on the MS-COCO dataset. Then, the output layers of the pretrained network were trimmed to have only two class outputs (sugar beet and potato) instead of the 80 class outputs of the MS-COCO dataset. Using the MS-COCO weights as initialization, we trained the C and D detection models on our sugar beet and potato dataset, as described in Section 2.1. Applying transfer learning to the CD-model required an additional step due to the different dimensions of the CD-model. The model pretrained on MS-COCO had kernels with a 3 × 3 × 3 dimension in the first layer of the network, whereas the CD-model had 3 × 3 × 6 kernels in the first layer because of the six-channel input images. To match this dimension, the three additional channels in the kernels in the first layer of the CD-model were initialized with a copy of the first three channels of the kernel from the pretrained model. 

### 2.3. Experiments

Three experiments were conducted to develop and evaluate two new detection models, namely the D- and CD-model, along with the reference C-model, which represents the current state-of-the-art. Experiments 1 and 2, outlined in Section 2.3.1 and Section 2.3.2, were conducted to identify the best data augmentation and transfer learning approaches for all three models, as well as to examine the effects of transfer learning and data augmentation on the novel D- and CD- models, which integrate depth data as input. Lastly, experiment 3 evaluated the performance of the three models in multiple vegetation densities and investigated the usefulness of the depth data for improving plant detection in dense scenes. 

For all three experiments, we used the all-point-interpolated average precision (AP@0.5) with an intersection-over-union threshold of 0.5 as the evaluation metric [47]. The AP is the precision averaged over all recall values. Furthermore, the mean AP@0.5 (mAP@0.5) was used. The mAP is the AP averaged over all object classes (sugar beets and potatoes). The AP@0.5 and mAP@0.5 are in the range of 0 to 1, where a score of 1 represents a perfect detection. 

Because the AP is a score between 0 and 1, it is by definition not normally distributed. Therefore, the non-parametric Wilcoxon rank test was used to analyze the significance of the differences between the results of the experiments. To run the Wilcoxon test, we computed an AP for each experiment, each class (sugar beet or potato), and each testing image. These AP values were then used in a two-sided Wilcoxon rank test to compute the *p* values. In our research we considered, *p* > 0.05 as an insignificant difference, *p* < 0.05 as a weak difference, and *p* < 0.01 as a strong difference. 

#### 2.3.1. Experiment 1—Data Augmentation

To assess the effects of the data-augmentation schemes on all three detection models, four different data augmentation schemes were evaluated. The schemes were: 1. no data augmentation, 2. geometric data augmentation, 3. spectral data augmentation, and 4. geometric and spectral data augmentation. The data augmentation process and its parameters were described in Section 2.2.3. During training, the transfer learning strategy was used, as described in Section 2.2.4.

It was expected that using geometric data augmentation would be equally beneficial for both the baseline model using color data and the models using depth data. This is because geometric transformations, such as rotations and flips, can occur for both color and depth data. Furthermore, because the spectral data augmentation was optimized for RGB color images, we expected the spectral data augmentation to be effective for the C-model using RGB color data. However, its effect was expected to be limited when applied to the D- and CD-model. 

#### 2.3.2. Experiment 2—Transfer Learning

In this experiment, we evaluated the effect of transfer learning of model weights trained on the MS-COCO dataset on the performance of our three detection models. Each of the models was trained twice, once with weights randomly initialized and once with weights pre-trained on the MS-COCO dataset, as described in Section 2.2.4. During training, geometric data augmentation was used, as described in Section 2.2.3.

We expected that transfer learning from the MS-COCO dataset would not be as effective for the D- and CD-model, which use depth data as the input, compared to the C-model, which only uses RGB color data. The reason for this is that the MS-COCO dataset does not contain any depth data. Therefore, the pre-trained weights might not be optimized for the type of data that the D- and CD-model are processing.

#### 2.3.3. Experiment 3—Depth Data in Dense Scenes

In this experiment, we evaluated the potential added value of depth information for plant detection in dense and occluded scenes. We evaluated the C-, D-, and CD-models using the best combination of transfer learning and data augmentation found in experiments 1 and 2. We evaluated these models using an independent test set that contained data with multiple vegetation densities, ranging from 0–10% to 80–90% vegetation coverage. 

It was expected that the addition of the depth data would help identify individual plants in dense and occluded scenes. Therefore, we expected that the CD-model, using both color and depth data, would outperform the other models, especially in dense scenes with high vegetation densities.

## 3. Results

In Section 3.1, the results of the data augmentation experiment are shown, and in Section 3.2, the results of the transfer learning experiment are shown. Finally, in Section 3.3, the detection performance of the three models for different vegetation densities is demonstrated. 

### 3.1. Experiment 1—Data Augmentation 

In Figure 5, the detection performance on the validation set of all the models as a result of the different data augmentation schemes is shown. For all the models, the performance on the potato class was lower than on the sugar beet class. This lower performance on the potato class can be explained by two factors. Firstly, there was a class imbalance, with 12,754 sugar beet annotations and 1427 potato annotations in the total training set, resulting in a ratio of approximately 1:9. Because of this imbalance, the algorithm focused more on the prevalent sugar beet class. Secondly, the potatoes exhibited greater variation in their appearance. The sugar beets were all sowed on the same date in the same depth within a strict pattern, ensuring homogeneous growth. Volunteer potatoes, on the other hand, are leftover tubers from previous years and therefore have no specific sowing date or sowing depth, resulting in a more heterogeneous growth [10]. 

When comparing the different models, overall, the results showed that the D-model consistently scored lower than the C-model (baseline model) on both the sugar beet and potato classes across all forms of data augmentation. This is likely because the depth data containes some level of discriminative features, but these were not as powerful as the features of the color data. On the other hand, the novel CD-model achieved similar results to the C-model, suggesting that the combination of color and depth data does not necessarily improve performance compared to color data alone. In the following section, a more comprehensive comparison of the models will be presented. Figure 6 displays the statistical significance of the performance difference between the models under various data augmentation schemes.

For the baseline model (the C-model), spectral data augmentation had no significant effect on either the sugar beet or the potato class. Geometric data augmentation had a strong positive effect, yielding an improvement of 0.03 for the sugar beet class and 0.20 for the potato class. When both data augmentation methods were used, no significant difference in the performance compared to using only geometric data augmentation was found.

On the other hand, for the D-model, spectral data augmentation had a significant negative effect, reducing the performance on the potato class by 0.02. This was likely because spectral data augmentation applied to the color-coded depth images caused changes to the color data that made it fall outside of the used color scheme, resulting in variation in the color-encoded depth images that was not representative of the original unaltered data. Geometric data augmentation, on the other hand, had a strong positive effect, improving the model’s performance on the sugar beet class by 0.03 and on the potato class by 0.28. This increase in performance was even larger than the increase in performance reported for the C-model. 

Similarly, for the CD-model, spectral data augmentation had a negative effect on performance. This is similar to the case of the D-model. Geometric data augmentation, on the other hand, had a strong significant positive effect, improving the performance for the sugar beet class by 0.03 and for the potato class by 0.35. This increase in performance was larger than the increase in performance reported for the C-model, indicating that, due to the different types of input data used, the CD-model requires more training data than the C-model. Using both spectral and geometric data resulted in a worse performance than using geometric data augmentation alone. 

Overall, geometric data augmentation had a significant positive effect on the performance of all the models across all the classes, while spectral data augmentation had a small, negative effect on the performance of all the models across all classes. The addition of both data augmentation methods never resulted in a performance that was significantly better than using geometric data augmentation alone.

### 3.2. Experiment 2—Transfer Learning

In Figure 7, the detection performance on the validation set of the C-, D-, and CD-models with and without transfer learning is shown. Applying transfer learning yielded improvements across all three detection models. For the sugar beet class, the performance increase after applying transfer learning was 0.02 AP for the C-model, 0.02 AP for the D-model, and 0.03 AP for the CD-model. For the potato class, the increase was 0.12 AP for the C-model, 0.14 AP for the D-model, and 0.13 AP for the CD-model. Overall, transfer learning was equally or more effective for the D- and CD-model than for the C-model. 

It is remarkable that transfer learning benefitted the models that included the depth information (the D- and CD-model) to the same extent as it benefitted the color model (C-model), despite pretraining being performed on the MS-COCO dataset, which only contained RGB-color images and did not contain color-encoded depth images. Therefore, it was expected that during pretraining, powerful color features would be learned that would be more beneficial for the C-model, which used color data. However, our experiments revealed that transfer learning was more effective for the D and CD-models, which used depth data. The authors suggest two reasons for this finding. First, the jet color encoding of the depth images encoded depth features in color space, making them more similar to the features that were pretrained on the MS-COCO dataset. Second, the pre-trained model used geometric features such as contours and edges that were color-agnostic and could be effectively transferred from the models using RGB-color images to the models using jet-encoded depth images.

### 3.3. Experiment 3—Depth Data in Dense Scenes

In Figure 8, the detection performance of the C-, D-, and CD-model applied to different vegetation densities is shown. The D-model performed worst for both classes, while the C- and CD-model performed better, with a performance above 0.90 AP for the sugar beet class across all vegetation densities. The detection performance for the potato class showed more variation for the different levels of vegetation densities. 

At low vegetation densities of 0–10%, both the C- and CD-model achieved a relatively low performance of approximately 0.70 AP. However, the detection performance of both models increased with an increase in vegetation density, achieving a peak performance of 0.95 AP for scenes with a vegetation density of 30–50%. As the vegetation density further increased, the performance gradually decreased to 0.70 AP for the C-model and 0.55 AP for the CD-model at an 80–90% vegetation density.

This trend can be explained based on the plant growth stages. In scenes with a low vegetation density of 0–10%, plants were small and showed limited differentiating features, making it difficult to correctly localize and classify the plants. An example of these scenes and the detection performances of the three detection models are given in Table 5, column one. Scenes with a vegetation coverage of 40% to 50% contained larger plants with more discriminative features but still had limited overlap and occlusion, making it easier to localize and classify the different plants. See Table 5, column two for an example. At a higher vegetation density of 80–90%, occlusion increased and identifying individual plants became more difficult, as shown Table 5, column three. In this scene, it became harder to separate the different sugar beet plants, and the potato plants were mostly occluded by the sugar beet plants. 

It was expected that the addition of depth data would be beneficial for distinguishing individual plants in occluded scenes [16,17]. Therefore, the CD-model was expected to outperform the C-model in scenes with high vegetation densities. However, our results showed the contrary, as the C-model outperformed the CD-model, especially in these dense scenes. This difference in performance was particularly evident for the potato class. The possible reasons for the mismatch of our expectations are elaborately discussed in Section 4.3.

## 4. Discussion

In Section 4.1, the results of the data augmentation experiments are discussed, and in Section 4.2, the results of the transfer learning experiment are discussed. Finally, in Section 4.3, the added value of depth data for dense vegetation is discussed. 

### 4.1. Data Augmentation 

Spectral data augmentation is a state-of-the-art method to improve the detection performance of color-based detection models. Therefore, we expected that spectral data augmentation would positively affect the performance of the reference model (C-model), which used color images for its detection, and the CD-model, which used both color and depth images for its detection. However, spectral data augmentation had no positive effect on any of the models. Similar results are reported by Blok et al. [25], who found that spectral data augmentation did not affect the performance of their broccoli detection model. It must be noted that our entire dataset is focused on sugar beets and potatoes in an agricultural environment; plants that have a quite narrow spectral range. The effects of data augmentation can be highly dependent on the object class, and data augmentation strategies that are beneficial for one class can be disastrous for another class [48]. For other detection tasks in the agricultural domain with objects that show more spectral variation, spectral data augmentation might be more beneficial. 

Geometric data augmentation, on the other hand, had a positive effect on all the detection models, especially on the D- and CD-model. For the D- and CD-model, applying geometric data augmentation yielded an increase of 0.28 and 0.35 AP for the underrepresented potato class, whereas using geometric data augmentation only improved the performance of the C-model by 0.20 AP for the same class. These findings demonstrate that geometric data augmentation is an effective strategy to improve the performance of a plant detection model using depth and/or color data as the input. 

As dealing with variation is a significant challenge in the practical application of autonomous weeding robots, it is important to further improve the generalization of the detection models. This can potentially be achieved through agricultural-specific data-augmentation strategies that better mimic the variations typically present in agriculture. Such data augmentation should therefore focus on creating variation in lighting, strong shadows, growth stages of both crops and weeds, soil type, excessive overlap, and the presence of different weed species [9,10]. 

### 4.2. Transfer Learning 

We hypothesized that applying transfer learning from the MS-COCO dataset to our three models would be the most beneficial for the C-model due to the similarity in the input data and less effective for the D- and the CD-model due to the different input data. However, our results showed that applying transfer learning from MS-COCO yielded a significant increase in performance for all three models, regardless of their input data used. We argue that the effectiveness of transfer learning for the models using depth data was caused by two reasons. The first reason is the usage of jet color encoding of the depth images, which encoded depth features in color space, making the features from the depth image more similar to the features pretrained on the MS-COCO dataset [33]. The second reason is that, in addition to color features, the trained models likely learned geometric features based on the edges and contours of the plants and leaves [49]. These geometric features are color agnostic and present in the color images as well as in the jet-encoded depth images. Hence, these features can be effectively transferred from models trained on RGB color images to models using jet-encoded depth images. 

These results contradict the current literature. Gupta et al. [50], Schwarz et al. [51], and Song et al. [52] claimed that large datasets with depth data are needed to effectively apply transfer learning to a deep learning model using depth data. However, compared to this literature, we used a different representation of the depth data. We used the jet color encoding scheme, as suggested by Eitel et al. [33], resulting in more similarity between the RGB images of MS-COCO and the depth images from our dataset.

Although the results were taken from a specific agricultural use case, we expect that the conclusions will also apply to the detection of other species of plants grown in arable fields observed from a top-down view, as these have similar image features. Therefore, we posit that pretraining an RGB-D-based plant detection model on the MS-COCO dataset is effective and that pretraining on a large-scale RGB-D dataset is not needed. 

### 4.3. Depth Data in Dense Scenes

We expected that the addition of depth data would improve the detection performance in scenarios with overlap and occlusion because in the past, using handcrafted algorithms, depth data were used to improve plant detection scenes [16,17]. And, more recently, Gené-Mola et al. [24] demonstrated the effectiveness of depth data for the detection of apples in dense scenes. However, our findings demonstrate that current state-of-the-art plant-detection algorithms do not benefit from the addition of depth data for the detection of sugar beet and potato plants. On the contrary, their performance is even slightly lower, especially for higher vegetation densities.

This unexpected result might be explained by two reasons. First, in the past, using handcrafted algorithms, depth information was a strong feature used to segment plants from the background and to discriminate between crops and weeds of different heights [16,17]. Using handcrafted algorithms, the amount of information that could be extracted from images was limited by the expert knowledge and available time of the algorithm designer, and depth was an easy-to-use feature. Current state-of-the-art detection methods are much better at extracting complex and reliable features from images because machine learning algorithms can exploit more features in the training data than a human expert can. Our results showed better results for the C-model than for the D- and CD-model, especially for the more complex potato plants, indicating that more powerful features could be extracted from the color images than from the depth images. 

A second reason is that there seems to be a high redundancy in the relevant information available about the plants in the color and in the depth images. Both types of images are in fact correlated, as the depth data are derived from two color images using a stereo vision algorithm. Important geometrical features, such as edges and contours, are therefore present in both the color image and the depth data. Furthermore, if an object is occluded in the color image, the object will also be occluded in the depth data, and if an area is over- or under-illuminated in the color image, its corresponding depth data will also be of a lower quality. Therefore, the depth data provides only a limited amount of additional information for our detection model applied to our use case of volunteer potato detection in sugar beet fields. Using different depth data encoding techniques, such as surface normals or a combination of the height above ground, horizontal disparity, and the angle with the gravity (HHA encoding) [53], will still maintain this correlation and therefore not increase the information present in the depth data. However, the usage of other depth sensors, such as lidar or multi-view stereo vision, could potentially yield higher-quality depth data, providing unique features that are not present in a single color image and potentially improving detection performance. 

In an extensive literature review on the use of depth data for agricultural applications, Kurtser and Lowry [23] identified only one paper in which a combination of color and depth data was used for object detection [24]. They dealt with the use case of apple detection. In their research, data was collected using a KinectV2 consisting of a color camera and an active time-of-flight infrared camera. Their data were collected at night, and the scene was illuminated using direct LED lighting without a diffuser, resulting in some over- an under-exposed apples. In their study, the addition of both depth and infrared data improved the detection of over- and under-illuminated apples. This suggests that the depth and infrared data only provided useful information for areas where color information was missing. And that similar gains might have been achieved by improving the color camera quality and lighting setup. 

We expect that the marginal effects of depth data on volunteer potato detection will generalize to the broader realm of plant detection in arable farming systems, unless the plant of interest has discriminative 3D characteristics that are not visible in its 2D projection. For instance, when 3D leaf curvature is a key feature, the addition of depth data might provide an improvement over using only color images. Additionally, depth data might offer added value in scenarios where color information is limited, such as under poor lighting conditions. For tasks other than plant detection, the addition of 3D information is indispensable, such as for biomass estimation [54] and plant length measurements [55].

## 5. Conclusions

In our study, we have introduced two novel plant detection models, the D-model, which uses depth data as the input, and the CD-model, which uses both color and depth data as inputs. We investigated the effects of data augmentation and transfer learning on these new models and investigated the potential of these models for plant detection in scenes with different vegetation densities. As a reference model, the C-model was used, a state-of-the-art detection model, using conventional color images as the input. 

It was hypothesized that the performance of the reference C-model and the novel D- and CD-model could be improved by training them using geometric data augmentation, and that the performance of the C- and CD-model could be further improved by adding spectral data augmentation. Geometric data augmentation indeed had a positive effect on all three detection models, especially for the underrepresented potato class, yielding an improvement of 0.20 AP for the C-model and improvements of 0.28 and 0.35 for the D- and CD-model, respectively. Spectral data augmentation, on the other hand, did not improve the performance of any of the models. We claim that the spectral data augmentation strategies are domain-specific. To further improve the effectiveness of plant detection models, agricultural-specific data augmentation strategies should be developed, in which excessive overlap, strong shadows, and the presence of a variety of weeds should be added.

When applying transfer learning from the MS-COCO dataset to our three models, it was expected that, due to the similarity in input data, the C-model would benefit more from transfer learning than the D- and CD-models. However, it was found that transfer learning yielded an equally large gain of approximately 0.02 AP for the sugar beet class and 0.13 AP for the potato class for all the models. Therefore, we conclude that pretraining a plant detection model for arable crops on the MS-COCO dataset was effective for the novel D- and CD-model, and that pretraining on a large-scale depth dataset is not needed.

Finally, it was expected that, compared to the C-model, the addition of depth data in the CD-model would improve the plant detection performance, especially in scenes with high vegetation densities. This increase was expected because, in the past, using handcrafted models, depth data were often used to improve plant detection in dense scenes. However, our results show the contrary: the addition of depth data did not improve—but rather worsened—the performance of the CD-model, especially at higher vegetation densities. The difference between the literature and our research is that, instead of handcrafted detection models, we used a state-of-the-art deep learning-based plant detection model. Deep learning models can extract powerful features from color images alone, which makes the features from depth images acquired using a stereo camera system less valuable. Therefore, we conclude that there is no added benefit to adding depth data for the detection of sugar beet and potato plants, not even in highly occluded scenarios. Whether depth data have benefits in other arable systems or other tasks remains a question for future work.

## Figures and Tables

**Figure 1 sensors-24-01942-f001:**
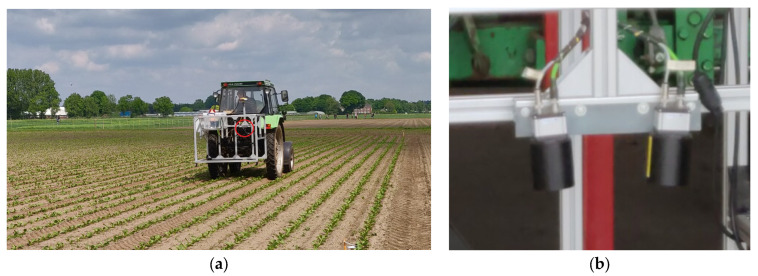
Image acquisition setup. (**a**) Image acquisition setup mounted behind a tractor in the field. Camera indicated by the red circle; (**b**) close-up of the stereo camera.

**Figure 2 sensors-24-01942-f002:**
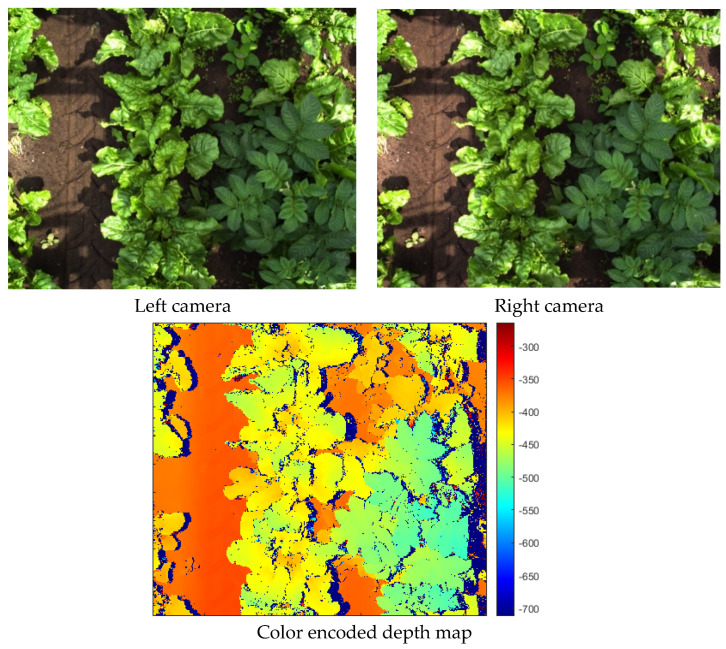
Rectified color images and color-encoded depth map with a color bar showing the disparity in pixels corresponding to each color. Note that due to direct natural sunlight, strong shadows are present in the color images and that the depth map does not show variations in illumination. Note that near the edges of the plant leaves, the depth values are missing because these areas are occluded in one of the images of the image pair.

**Figure 3 sensors-24-01942-f003:**
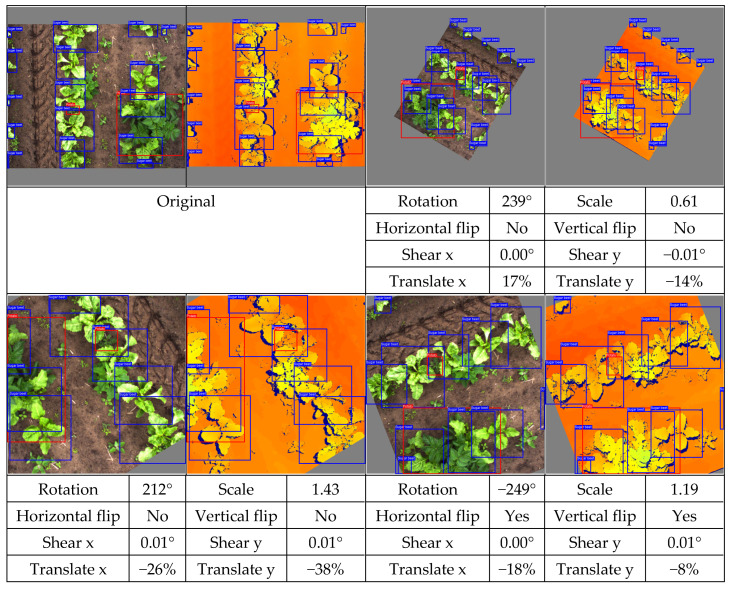
Examples of random geometric data augmentation. Each of these example images is derived from the same original image. One can see changes in rotation, translations, scale, and flips. The effects of the shear transformation are limited. Note that the color-encoded disparity map and the color image received the exact same transformation and are therefore still aligned.

**Figure 4 sensors-24-01942-f004:**
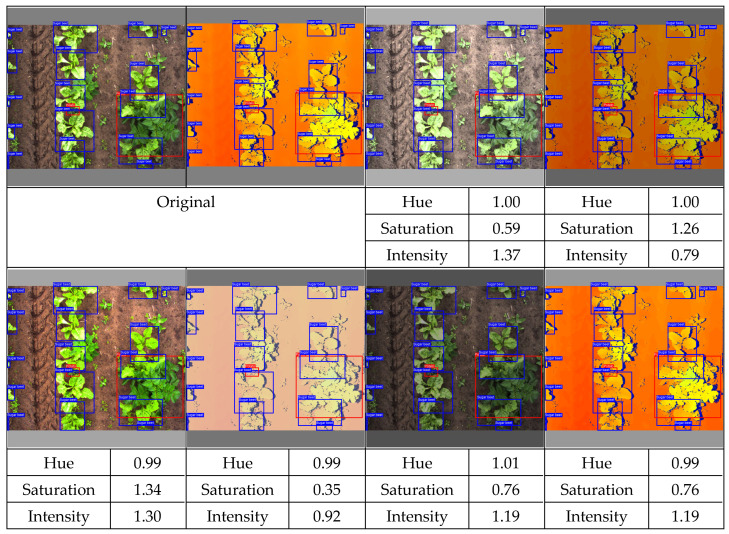
Examples of spectral data augmentation. Note that the depth map and the color image did not receive the exact same spectral transformation. The brightness on the depth map might be increased, while the brightness on the color image might be decreased.

**Figure 5 sensors-24-01942-f005:**
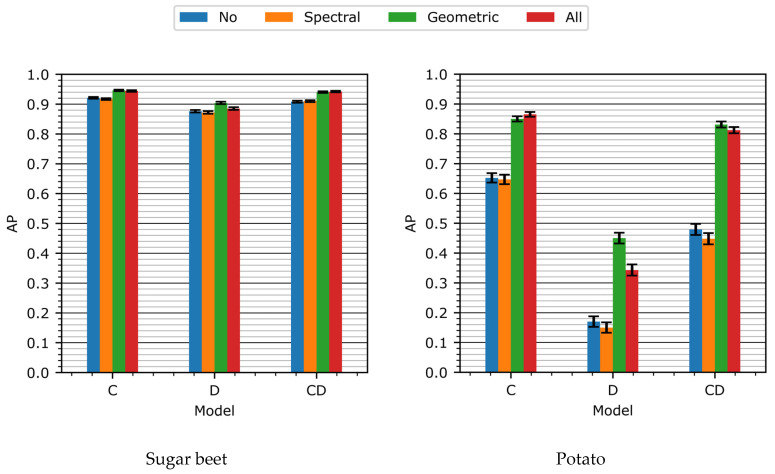
Detection performance as result of different data augmentation schemes. In the left graph, the model’s performance for the sugar beet class is shown, and in the right graph, the model’s performance for the potato class is shown. The error bars represent the standard error of the mean, computed over all test samples.

**Figure 6 sensors-24-01942-f006:**
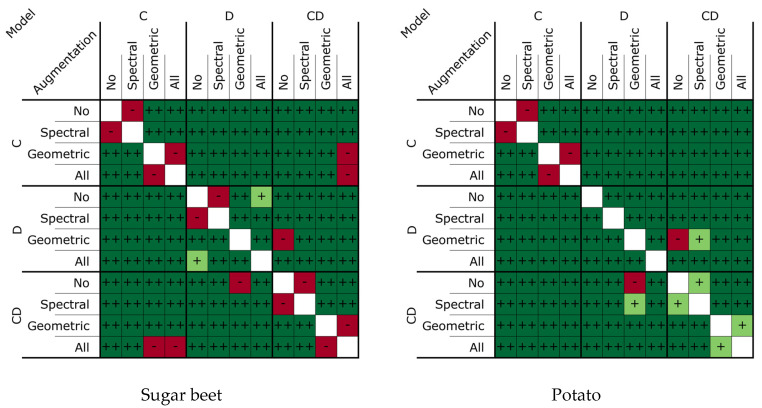
Significance of the difference between the different model instances tested using a two-sided Wilcoxon test. In the left figure, the results of the sugar beet class are shown, and in the right figure, the results of the potato class are shown. Insignificant differences (*p* > 0.05) are denoted with ‘-’. Weak differences (*p* < 0.05) are denoted with ‘+,’ and strong differences (*p* < 0.01) are denoted with ‘++’. The majority of the cases had a strong difference.

**Figure 7 sensors-24-01942-f007:**
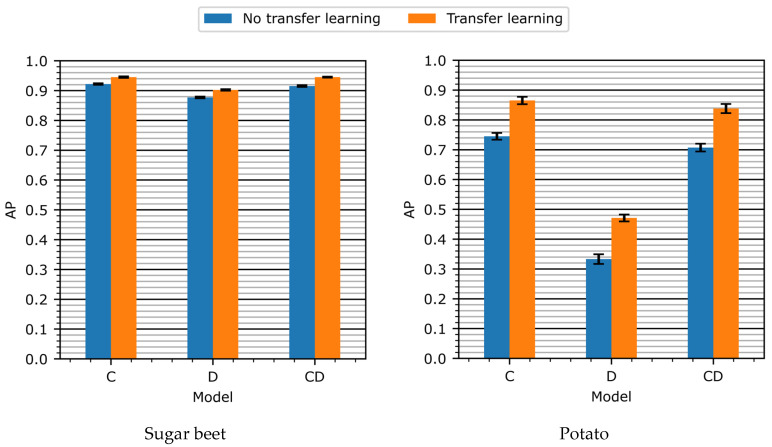
Detection performance for the sugar beet and the potato class as result of the usage of transfer learning. The error bars represent the standard error of the mean, computed across all test samples.

**Figure 8 sensors-24-01942-f008:**
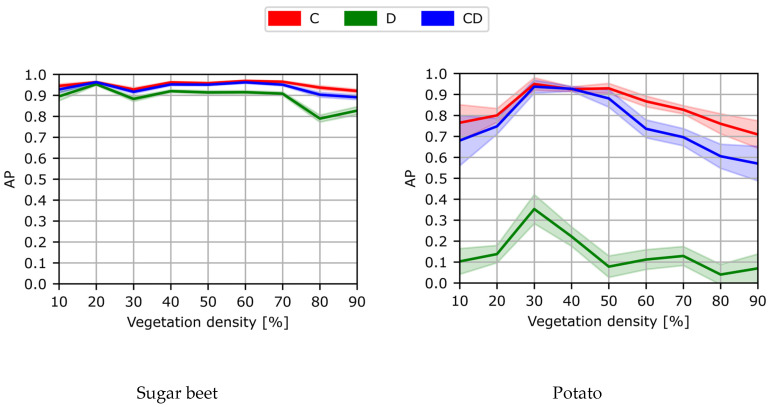
The detection performance of the C-, D-, and CD-model for different vegetation densities. On the y-axis, the detection performance is expressed as the AP, and on the x-axis, the vegetation density is given as the percentage of the image covered in green pixels. Performance is computed for every 10% vegetation density category. In the left figure, the performance for the sugar beet class is shown, and the right figure shows the performance for the potato class. The transparent areas represent the standard error of the mean.

**Table 1 sensors-24-01942-t001:** The training and validation datasets and their meta-information, consisting of the acquisition dates; the number of images acquired during each acquisition run; the number of annotated sugar beets; the number of annotated potatoes, the growth stage of the sugar beets; the wind speed at 2 m above the ground, recorded by a nearby weather station “De Veenkampen;” and a qualitative assessment of the sunlight conditions.

Field	Date	Images	Sugar Beets	Potatoes	Wind Speed [m/s]	Sunlight
Training	22 May 2019	201	2826	399	3.1	Direct
	30 May 2019	240	3747	399	4.2	Diffuse
	3 June 2019	244	3742	381	3.5	Diffuse
	6 June 2019	180	2439	248	3.4	Direct
Validation	22 May 2019	127	2237	215	3.1	Direct
	30 May 2019	114	2008	219	4.2	Diffuse
	3 June 2019	109	1876	218	3.5	Diffuse
	6 June 2019	90	1345	116	3.4	Direct

**Table 2 sensors-24-01942-t002:** The test dataset and its meta-information, consisting of the vegetation density, the number of images acquired for each vegetation density, the number of annotated sugar beets, and the number of annotated potatoes.

Field	Vegetation Density	Images	Sugar Beets	Potatoes
Test	0–10%	11	212	20
	10–20%	59	1060	94
	20–30%	28	569	61
	30–40%	60	1150	138
	40–50%	42	852	104
	50–60%	60	1120	102
	60–70%	67	1230	111
	70–80%	44	709	65
	80–90%	23	361	30

**Table 3 sensors-24-01942-t003:** Geometric data augmentation parameters used in this study.

Augmentation	Range
Rotation	+/−360°
Translation (vertical and horizontal)	+/−10% of the image dimensions
Scale	+/−50% of the image dimensions
Shear (vertical and horizontal)	+/−1°
Vertical flip	50% probability
Horizontal flip	50% probability

**Table 4 sensors-24-01942-t004:** Spectral data augmentation parameters.

HSV hue	+/−1.5%
HSV saturation	+/−70%
HSV intensity	+/−40%

**Table 5 sensors-24-01942-t005:** Detection performances of the C-, D-, and CD-model across three different vegetation densities. The three columns show vegetation densities of 0–10%, 40–50%, and 80–90%. For each vegetation density, one scene is selected, and the detections on the same scene are shown for each detection model. Row 1 shows the detections of the C-model visualized on the color images, row 2 shows the detections of the D-model, visualized both on the color and on the depth images, and row 3 shows the detections of the CD-model, visualized on both the color and the depth images.

Legend	 Ground Truth Sugar Beet	 Detection Sugar Beet	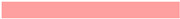 Ground Truth Potato	 Detection Potato
	0–10%	40–50%	80–90%
1. C	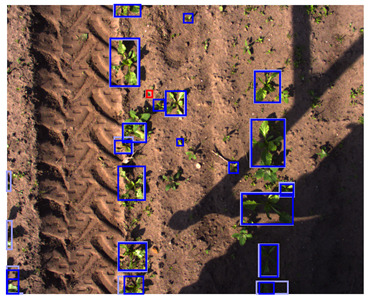	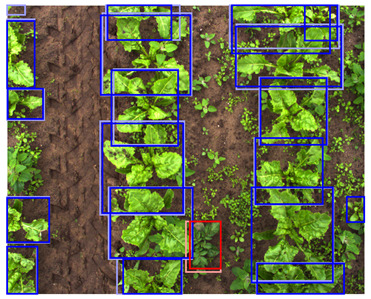	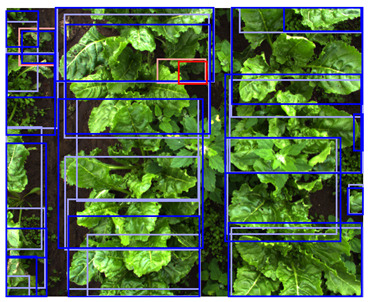
2. D	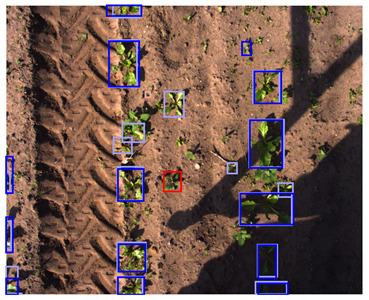	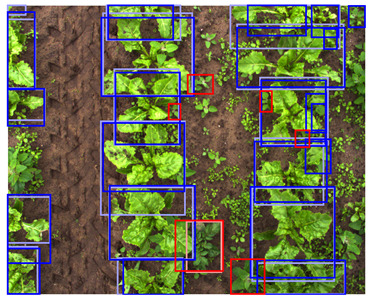	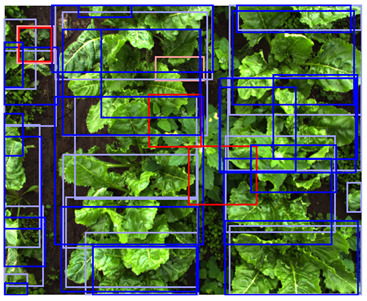
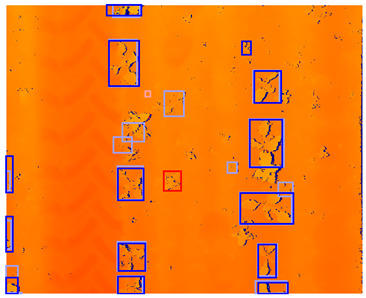	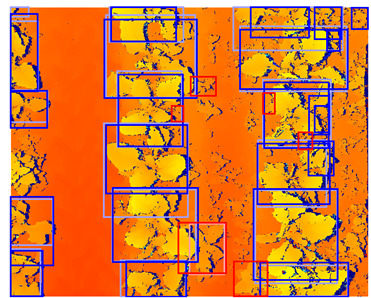	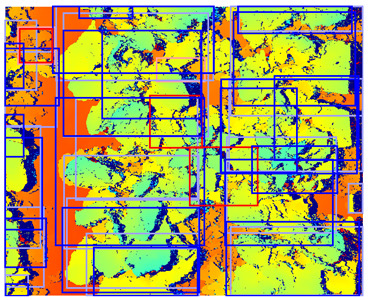
3. CD	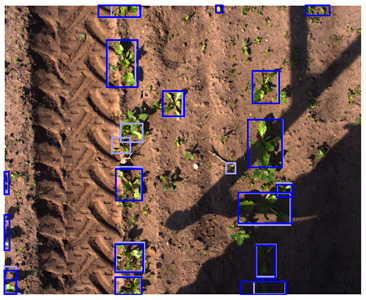	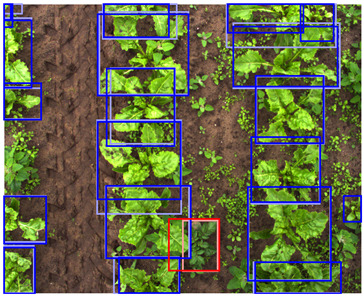	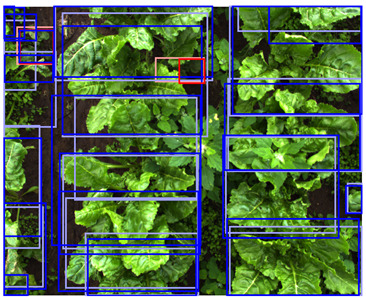
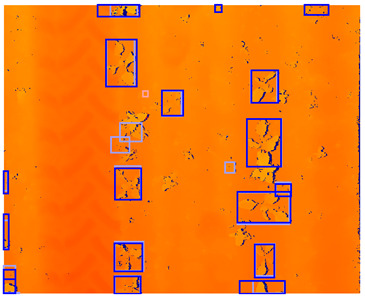	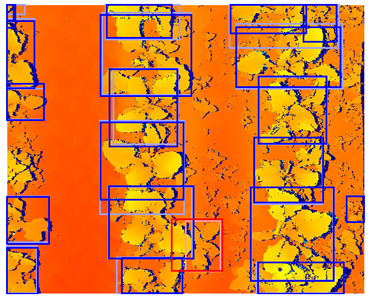	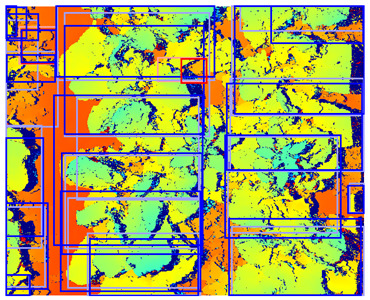

## Data Availability

The image data used for this study will be made available at http://doi.org/tbdt.

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
