# Peer review of "Stereo Vision for Plant Detection in Dense Scenes"

_sensors, 2024, doi:10.3390/s24061942_

Round 1
Reviewer 1 Report
Comments and Suggestions for Authors
Good job! Please check my comments below.
1. Abstract: can you provide the quantitative performance evaluation? For example, how much is the improvement of “combining color encoded depth data, with geometric data augmentation and transfer learning can improve an RGB-D detection model”?
2. The title and abstract are not a good match, you mentioned “stereo vision” but not shown in abstract.
3. What does the importance of detecting weed in dense agricultural fields? It might be too late to do management.
4. Lines 117-125, 194-199, 335… what is “section 0”?
5. Section 2.1.1, can you add a picture to show the cameras set up? Perhaps zoom in the current figure 1, and get detailed information of cameras. From the current figure 1, I cannot see the cameras.
6. What’s the weather condition during images collection? Is it sunny or cloudy? Windy or windless?
7. Can you provide a picture of the checkboard calibration plate you are using?
8. Line 149, what is “semi-global matching”?
9. What is the library, programming language, and hardware (e.g., CPU, GPU, memory) to perform this study (YOLO, transfer learning)?
10. What are the parameters and values of camera intrinsic and extrinsic?
11. Why potato performance lower than sugar beet performance?
Reviewer 2 Report
Comments and Suggestions for Authors
Comments paper “Stereo vision for plant detection in dense scenes.”
Ruigrok et al., 2024, Sensors 2600350
The paper describes two new models to improve the automated precision weed-control methods to discriminate between crops and weeds. The current state-of the art plant detection do not meet the criteria to detect weeds, especially in dense and occluded scenes. The former hand-crafted detection methods using color and depth are not satisfactory and the combination of these features is not widely used in deep-learning-based vision systems in agriculture. The paper now presents two new models, the D-model, using depth data, and the CD-model using both color and depth data as input and the C-model as reference. The main conclusion of the authors is that geometric data augmentation and transfer learning were equally effective for both the reference model and the two models using depth data. However, testing of the detection methods on a volunteer potato – sugar beet farming plot did not improve plant detection at high vegetation densities.
The paper is well written, the introduction and the sections Materials & Methods, Results and Discussion are extensively and thoroughly described. An important point is that the authors present a work in which the expectation that, compared to the C-model, the addition of depth data in the CD-model, would improve plant detection performance in scenes with high vegetation density, could not be realized at this moment. This is in my opinion important information for the community working in this field, in order to realize the limitations of techniques and data analysis.
However, the manuscript can be improved in such way that also not only the “super-expert” in this domain, but also less experienced people in the community, could profit of the analysis used and the results.
p.1, lines 36,37 and 39: explain the abbreviations- YOLO, SSD, Mask-RCNN and MS-COCO
p.2, line 54: explain RGB-D
p.6, line 180: explain “labelImg”
p.7, line 195: explain YOLOv5L, what is “5vL” standing for?
p.7, line 210: explain “mAP”
p.21, line 578: explain “HHA encoding”
correct every time “section 0”: lines 117, 118, 194, 195, 198, 335, 359, 360, 371, 372, 390, 391, 487, 490, 491.
Round 2
Reviewer 1 Report
Comments and Suggestions for Authors
1. Figure 1, can you add an arrow to indicate the position of the stereo camera in the tractor?
2. What is the data source for wind speed and sunlight condition? Is it quality controlled data?
3. How do you deal with the imbalanced dataset of potato and sugar beet? If you had two models, one for potato, another for sugar beet, “imbalance” is not a good term.
4. These two references have the same title:
“Ruigrok, T., Henten, E. J. Van, Kootstra, G. (2022). Improved generalization of a plant-detection model for precision weed control. Manuscript Submitted for Publication.
Ruigrok, T., van Henten, E. J., Kootstra, G. (2023). Improved generalization of a plant-detection model for precision weed control. Computers and Electronics in Agriculture, 204, 107554. https://doi.org/10.1016/j.compag.2022.107554”
5. Lack of full information of references:
“The Effects of Regularization and Data Augmentation are Class Dependent”, lack of conference name of this proceeding;
“Automatic Detection and Classification of Weed Seedlings under Natural Light Conditions”, lack of the university of this dissertation;
“Multimodal deep learning for robust RGB-D object recognition.”, from APA format it is 2015, September, instead of “2015-Decem”;
“Fawakherji, M., Potena, C., Prevedello, I., Pretto, A., Bloisi, D. D., Nardi, D. (2020). Data Augmentation Using GANs for Crop/Weed Segmentation in Precision Farming. CCTA 2020 - 4th IEEE Conference on Control Technology and Applications, (July 2021), 279–284. https://doi.org/10.1109/CCTA41146.2020.9206297”, is this one in 2020 or 2021?
“Microsoft COCO: Common Objects in Context”, lack of conference name of this proceeding;
Can you use full name of “CVPR”?
“CNN based Color and Thermal Image Fusion for Object Detection in Automated Driving”, lack of conference name for this proceeding.
